# Production and provenance of architectural glass from the Umayyad period

**Laura Ware Adlington**[1], **Markus Ritter**[2], **Nadine Schibille**[1]*

**1** IRAMAT-CEB, UMR5060, CNRS, Orléans, France, **2** Department of Art History, University of Vienna, Vienna, Austria

* nadine.schibille@cnrs-orleans.fr

**Data Availability Statement:** All relevant data are within the manuscript and its Supporting Information files.

**Funding:** The project received funding from the European Research Council (ERC) under the

## Abstract

A large assemblage (n = 307) of architectural glasses (tesserae and windows) from the early 8th-century Umayyad residential site at Khirbat al-Minya was analysed by laser ablation inductively coupled plasma mass spectrometry. Trace element patterns are essential to establish the provenance of the base glass, while the comparative evaluation of the colouring and opacifying additives allow us to advance a production model for the manufacture of glass mosaic tesserae during the early Islamic period. The primary glass types are Levantine I and Egypt 1a, as well as a few older, reused tesserae, and Mesopotamian plant ash glass used for amber-coloured window fragments. Chemical data revealed fundamental differences in the colouring and opacification technologies between the Egyptian and Levantine tesserae. Co-variations of lead and bismuth, and copper, tin and zinc in the Egypt 1a tesserae provide first evidence for the production of different mosaic colours in a single workshop, specialising in the manufacture of tesserae of different colours. No such trend is apparent in the Levantine samples. Red, cobalt blue and gold leaf tesserae were found to be exclusively made from a Levantine base glass, indicating that the generation of some colours may have been a specialised process. The same may apply to the amber-coloured window glass fragments of Mesopotamian origin that exhibit very unusual characteristics, combining elevated copper (2% CuO) with an excess in iron oxide (5% $Fe_2O_3$). These findings have significant implications for the production model of strongly coloured glass and the exploitation of resources during the early Islamic period.

## 1 Introduction

The late 7th and early 8th century was a transitional time for the glass-making industry of the Levant, one of the centres for the production of raw glass during the first millennium CE. Glass made with sand as the silica source and natron as the flux was produced in a centralised production system [1–3], in which a few primary workshops produced large amounts of raw glass. This raw glass was then distributed widely to secondary workshops across the empire, where shaping and colouring took place. Large tank furnaces for primary production of natron glass have been found in the Levant (4th– 8th century CE) and in Egypt (mostly 1st– 3rd century CE) [4–6]. Meanwhile in Mesopotamia, glass was made with plant ash as the principal flux throughout the late antique period [7–9].

European Union's Horizon 2020 research and innovation programme (grant agreement No. 647315 to NS). URL: https://ec.europa.eu/programmes/horizon2020/en/h2020-section/european-research-council The funders had no role in study design, data collection and analysis, decision to publish, or preparation of the manuscript.

**Competing interests:** The authors have declared that no competing interests exist.

Centuries of glass-making at Apollonia (Sozusa, Arsuf), 15 km north of Tel Aviv, ceased sometime during the late 7th or early 8th century and moved to nearby Bet Eliʿezer (Hadera, Hudayra) [10]. This shift has been connected to the political, social and cultural changes occurring at the time under the Umayyad caliphate (661–750 CE), with possible links to the increased building works emblematic of this period [10–12]. However, glass assemblages and especially architectural glass (such as mosaic tesserae and window glass) of this transitional period are not well characterised, with recent studies constrained by a very limited sample size [13, 14]. The abundant vitreous material recovered from the palatial residence of Khirbat al-Minya (Israel) provides a nuanced and complex picture of the supply of glass for architectural decorations at a very precise moment in time.

Khirbat al-Minya is an Umayyad residential complex located on a fertile plain at the north shore of Lake Tiberias in modern day Israel. The building has been excavated in several seasons in 1932–39 [15–20], in 1959 [21], and some soundings have been made in recent years [22, 23]. Some attention has been paid to the history of the settlement at the site [24], and finds of the unglazed and glazed ceramics [25, 26] as well as glass bracelets have been studied [27]. Recently, the archaeological record from the excavations has been summarised and reviewed, adding hitherto unpublished material including a record of the coins and an examination of a building inscription as part of a study of the architecture and decoration of the Umayyad structures [28]. The inscription names the caliph al-Walīd as the patron, indicating a princely association and connecting the building to either al-Walīd I (r. 705–715 CE) or al-Walīd II (r. 743–744 CE). The evidence of the coins favours the earlier dating [28, 29].

In the Umayyad building, which has four wings around a colonnaded courtyard, a monumental portal on the east side and two staircases indicating an upper storey, a main construction phase and a secondary phase have been identified. The latter phase saw the addition of a ramp in the southeast portico leading to the upper storey and some other minor changes. A mosque in the southeast corner was part of the main phase of construction and the presence of a prayer niche dates it to after 707 CE. The mosque and one room in the north wing appear to have remained unfinished, but the presence of coins and pottery indicate an Umayyad use of the building [28]. The site was severely affected and partially destroyed by the earthquake in 749 CE [28, 30].

Later occupations of the site involved structural and functional changes. For example, the portal was repaired, and the installation of a large kiln on the Umayyad floor in the south wing indicates a change in the structure and a non-residential function. The east and west wings were modified, using smaller basalt stones instead of the large limestone blocks employed in the Umayyad masonry. In the east wing, a single long hall with a middle row of supports was erected over the original suite of separate rooms [16, 17, 28]. Various types of unglazed and glazed ceramics date from the eighth to the 14th/15th centuries CE [25, 26]. The largest number of coin finds date from the Abbasid period up to the end of the 9th century, and again from the Zangid-Ayyubid times in the 11th to the 13th centuries. Fatimid and Crusader money from the late tenth to the late 12th centuries is mostly absent. The record continues with coins from the Mamluk period, until the beginning of the 16th century; there are no Ottoman coins [28]. A few finds of 4th- to 6th-century ceramics and coins outside the Umayyad building, 1–1.5 m below its floor level, indicate some pre-Islamic presence at the site [19, 28].

Large quantities of dislocated tesserae from wall mosaics and one of the largest assemblages of early Islamic window glass currently known have been recovered from various parts of the building and the courtyard. The largest quantity of loose tesserae and fragments of wall mosaic in mortar bedding were excavated on the floor of a three-aisled reception hall in the centre of the south wing. The tesserae include glass of various colours, with shades of red, blue, green, yellow, purple, black and gold leaf tesserae, as well as natural stone. The floor mosaics in a five-

room group (bayt) in the south wing are predominantly of stone but also contain some coloured glass. A deposit of fist-size stones identical in colour to the stone tesserae in the floor mosaics with marks and cut in such a way to produce tesserae, was found in a room adjacent to the mosque. One large deposit of glass tesserae, found on the floor in a room of the north wing without any attached fragments of mortar, appears to have been a repository for materials. Both deposits on the Umayyad floor level seem to indicate that work on the mosaics had remained unfinished. Additionally, fragments of glass possibly connected to mosaic production were found. In particular, a fragment of a glass slab, consistent with the glass tesserae in its thickness and green colour, appeared to have marks from impact and pinching at the edges, suggesting this could be material from which some of the glass tesserae may have been cut on site [28]. The window glasses were originally used in stucco transennae of different designs, of which fragments were found in various parts of the building. The window glass is generally well preserved, cut into different shapes with their edges intact, and includes various colours, mostly muted and cool hues. Neither the glass tesserae nor the window glasses can be related to the post-Umayyad phases of the building, which are reduced in scale and poorer in quality. They have recently been documented, visually described and discussed in relation to their architectural use [28].

In this study, we use the compositional characteristics of a large selection (n = 307) of the glass tesserae and window glasses from Khirbat al-Minya to advance a model of supply. This is the first time that such a substantial and exceptionally well-dated assemblage of architectural glass from an early Umayyad context has been studied in detail, using high resolution laser ablation inductively coupled plasma mass spectrometry (LA-ICP-MS). While the main features of the late antique glass industry are by now well understood, the organisation of the production and supply of strongly coloured glass mosaic tesserae are still largely unknown, mostly due to the limited number of samples from secured archaeological contexts that have been investigated [13, 14]. Our large-scale approach, which confirms the import of either finished (tesserae) or semi-finished (cakes) products from both Egypt and the Levant, yields unique insights into supply patterns and secondary working practices. Covariances between colouring agents further reveal common secondary production processes, suggesting that multiple mosaic colours were made at the same secondary workshop.

## 2 Methods

The architectural glass finds from Khirbat al-Minya nowadays housed in the Museum of Islamic Art in Berlin were chemically characterised with the official permission of the museum to study primary production (base glass types) and secondary glass working (colouring and opacifying). No further permits were required, and our study complies with all relevant regulations. There is no further repository information available, individual sample numbers were assigned for the purpose of this study and are accordingly archived in the Museum of Islamic Art in Berlin. 270 glass tesserae and 37 window glass pieces were selected for analysis by laser ablation inductively coupled plasma mass spectrometry (LA-ICP-MS). A subset of 60 tesserae and 14 window glass fragments (samples with the prefix "NS", see S1 Table) were sampled by removing a small amount of material, which was then embedded in epoxy resin with the cross-section exposed and polished to 1 μm using diamond paste. The remaining 210 tesserae and 23 window glass fragments were analysed after cleaning but without further sample preparation (samples with the prefix "MR").

The chemical analyses by LA-ICP-MS were carried out at the Centre Ernest Babelon (CEB) of IRAMAT in Orléans (France) using a Resonetics M50E excimer laser (ArF, 193 nm) equipped with a S155 ablation cell and a Thermo Fisher Scientific ELEMENT XR mass

spectrometer system. The mounted as well as the unprepared samples were ablated in spot-mode with a 5 mJ energy, a 10 Hz pulse frequency and a beam diameter typically set at 100 μm that was occasionally reduced when saturation of the signal caused by high concentrations of manganese and/or tin particles occurred. All samples were pre-ablated for 20 s to remove any surface alterations, acquisition time was set at 30 s. The ablated material is transported to the plasma torch by an argon/helium flow at an approximate rate of 1 L/min for Ar and 0.65 L/min for He. The ion signals in counts-per-second are recorded for 58 isotopes (from Li to U). A combination of internal ($^{28}$Si) and external standards (NIST SRM610, Corning glasses B, C, and D and an in-house APL1) were used for calibration. Quantitative data were calculated based on the procedures described by Gratuze [31]. Instrumental precision and accuracy were monitored throughout the analyses by measuring synthetic reference materials (NIST SRM612, Corning A, B, C and D) at regular intervals (S2 Table) [32, 33]. Analyses of NIST 612 generally resulted in a precision better than 5%, and accuracy better than ±5% with very few exceptions. The Corning analyses had better than 5% precision and most elements compare well with the Corning values (within about 5% accuracy) with some exceptions, in particular $P_2O_5$ in Corning A and C (which compare better with the Brill [34] values rather than the Wagner [35] values).

The cross-sectioned tesserae in resin blocks were carbon-coated and examined using a Zeiss EVO-25 scanning electron microscope coupled with an Oxford Instruments energy dispersive spectrometer (SEM-EDS) and Aztec software. Semi-quantitative chemical analyses were undertaken on different phases and inclusions for simple identification, with the SEM-EDS operating at a voltage of 20 kV, probe current of 1 nA, a working distance of 8.5 mm, and the time of analysis set to the collection of 750,000 x-ray counts with a deadtime of 40%. Corning A-C were analysed as secondary standards, with accuracy and precision better than 10% for all elements present in excess of 1 wt%. Secondary electron and backscattered electron images were thus taken.

## 3 Results

### 3.1 Base glass composition (primary production)

The chemical compositions of the tesserae and window fragments from Khirbat al-Minya (Table 1, S1 Table) are predominantly consistent with the use of a natron-type glass with characteristically low MgO, $K_2O$ and $P_2O_5$ contents as the base to which other materials were added for colouring and/or opacifying. Several of the tesserae, comprising the green and yellow samples, have compositions consistent with the combination of significant amounts of lead-tin oxide to a natron glass (with 6–32% PbO), which would have contributed to the generation of their colour (see next section), but also affect the absolute concentrations of elements in the glass. This problem is sometimes mitigated in other publications through the use of reduced compositions [34]. The reduced composition is a selection of base glass elements that are then normalised, leaving out all additives (colourants, opacifiers) while sometimes setting upper limits for some base glass elements affected by colourants (such as iron). In this paper, we generally favour element ratios over reduced compositions in our reporting and interpretation of the data as this does not require unnecessary transformation of the data.

Elements related to the silica source may be used to define widely recognised natron glass types in the literature, which are related to the primary production of glass (Fig 1A). About one-third of the analysed tesserae ($n = 102$) and half of the window glass fragments ($n = 18$) are comparable to the Egypt 1a type (Fig 1A and 1B), the production of which is dated to the first quarter of the 8[th] century CE [36]. These samples contain high alumina ($Al_2O_3/SiO_2 >$ 0.05) and high heavy mineral contents ($TiO_2/Al_2O_3 \cong 0.07$, 70–120 ppm Zr). Three window

**Table 1. Mean chemical compositions of mosaic tesserae and window glass from Khirbat al-Minya as measured by LA-ICP-MS divided according to primary glass type and colour.**

| Base glass | Colour | wt% | | | | | | | | | | | | | | ppm | | | | |
|---|---|---|---|---|---|---|---|---|---|---|---|---|---|---|---|---|---|---|---|---|
| | | Na$_2$O | MgO | Al$_2$O$_3$ | SiO$_2$ | P$_2$O$_5$ | Cl | K$_2$O | CaO | TiO$_2$ | MnO | Fe$_2$O$_3$ | CuO | SnO$_2$ | PbO | Li | Cr | Co | Sr | Zr |
| **TESSERAE** | | | | | | | | | | | | | | | | | | | | |
| Egypt 1a | Green (n = 31) | 15.6 | 0.58 | 3.31 | 61.4 | 0.16 | 0.94 | 0.47 | 3.00 | 0.24 | 0.08 | 1.02 | 1.22 | 1.05 | 10.7 | 2.89 | 35.2 | 8.81 | 204 | 84.8 |
| Egypt 1a | Olive (n = 7) | 16.3 | 0.61 | 3.52 | 67.5 | 0.45 | 0.99 | 0.54 | 3.61 | 0.27 | 0.04 | 1.08 | 0.07 | 0.62 | 4.32 | 2.99 | 39.3 | 4.84 | 213 | 99.3 |
| Egypt 1a | Purple (n = 24) | 17.2 | 0.73 | 3.70 | 69.2 | 0.64 | 0.96 | 0.49 | 4.28 | 0.27 | 1.11 | 1.16 | 0.04 | 0.01 | 0.06 | 2.53 | 38.0 | 67.2 | 306 | 94.6 |
| Egypt 1a | Turquoise (n = 37) | 16.9 | 0.65 | 3.62 | 68.1 | 0.87 | 0.99 | 0.53 | 4.26 | 0.27 | 0.23 | 1.14 | 1.50 | 0.10 | 0.57 | 3.06 | 38.8 | 18.6 | 241 | 93.9 |
| Egypt 1a | Yellow (n = 3) | 13.8 | 0.58 | 2.99 | 55.6 | 0.31 | 0.82 | 0.39 | 2.95 | 0.21 | 0.15 | 0.93 | 0.64 | 2.13 | 18.3 | 2.83 | 32.7 | 12.6 | 196 | 75.6 |
| Levantine | Aqua blue (n = 14) | 13.8 | 0.70 | 2.98 | 69.6 | 0.50 | 0.86 | 0.80 | 9.17 | 0.08 | 0.26 | 0.49 | 0.16 | 0.06 | 0.39 | 4.95 | 15.2 | 3.75 | 453 | 43.3 |
| Levantine | Black (n = 12) | 13.7 | 0.73 | 2.94 | 63.6 | 0.34 | 0.73 | 1.24 | 9.60 | 0.09 | 0.33 | 5.72 | 0.10 | 0.08 | 0.59 | 5.20 | 17.0 | 25.9 | 458 | 48.4 |
| Levantine | Blue (n = 22) | 14.0 | 0.73 | 3.26 | 67.9 | 0.22 | 0.79 | 0.87 | 10.2 | 0.09 | 0.31 | 0.86 | 0.12 | 0.01 | 0.43 | 4.04 | 18.1 | 465 | 491 | 45.8 |
| Levantine | Gold leaf (n = 10) | 13.9 | 0.74 | 2.95 | 67.0 | 0.20 | 0.74 | 0.84 | 10.1 | 0.10 | 2.53 | 0.73 | 0.01 | 0.00 | 0.02 | 4.95 | 23.5 | 7.07 | 476 | 50.0 |
| Levantine | Green (n = 21) | 12.2 | 0.57 | 2.64 | 58.2 | 0.14 | 0.72 | 0.74 | 7.93 | 0.07 | 0.12 | 0.45 | 0.95 | 1.63 | 13.5 | 4.00 | 12.4 | 8.32 | 396 | 38.8 |
| Levantine | Olive (n = 5) | 14.2 | 0.77 | 3.04 | 69.3 | 0.22 | 0.82 | 0.97 | 9.76 | 0.08 | 0.07 | 0.53 | 0.02 | 0.00 | 0.02 | 4.04 | 16.6 | 6.52 | 492 | 44.8 |
| Levantine | Red (n = 17) | 14.0 | 0.77 | 2.84 | 63.5 | 0.41 | 0.79 | 1.05 | 9.47 | 0.09 | 0.51 | 3.06 | 1.06 | 0.30 | 1.99 | 3.81 | 17.9 | 40.5 | 470 | 47.8 |
| Levantine | Turquoise (n = 17) | 14.1 | 0.74 | 2.82 | 68.7 | 0.22 | 0.79 | 0.79 | 8.31 | 0.08 | 0.08 | 0.50 | 1.55 | 0.17 | 0.87 | 4.07 | 12.7 | 3.59 | 448 | 43.8 |
| Levantine | Yellow (n = 21) | 10.6 | 0.50 | 2.37 | 54.0 | 0.09 | 0.66 | 0.50 | 6.29 | 0.06 | 0.09 | 0.37 | 0.21 | 2.91 | 21.3 | 3.69 | 10.4 | 3.68 | 325 | 33.6 |
| Foy 2.1 | Black (n = 4) | 16.8 | 0.88 | 2.45 | 65.8 | 0.16 | 0.85 | 0.74 | 8.81 | 0.13 | 2.26 | 0.86 | 0.02 | 0.00 | 0.07 | 5.85 | 17.2 | 93.6 | 571 | 69.4 |
| Foy 2.1 | Blue (n = 4) | 17.0 | 1.03 | 2.48 | 64.6 | 0.16 | 0.82 | 0.82 | 8.50 | 0.14 | 2.01 | 2.07 | 0.03 | 0.00 | 0.12 | 6.76 | 17.1 | 121 | 662 | 77.6 |
| Foy 2.1 | Gold leaf (n = 11) | 16.5 | 1.06 | 2.73 | 65.3 | 0.18 | 0.80 | 0.85 | 8.32 | 0.15 | 1.39 | 2.31 | 0.09 | 0.01 | 0.13 | 6.19 | 18.8 | 133 | 650 | 86.2 |
| **WINDOW GLASS** | | | | | | | | | | | | | | | | | | | | |
| Mesopotamian | Olive / Amber (n = 8) | 14.1 | 2.79 | 3.02 | 62.1 | 0.26 | 0.49 | 2.20 | 7.13 | 0.21 | 0.62 | 4.56 | 1.88 | 0.10 | 0.30 | 17.8 | 275 | 12.9 | 353 | 76.7 |
| Egypt 1a | Aqua (n = 2) | 17.8 | 0.76 | 3.99 | 71.0 | 0.08 | 1.02 | 0.51 | 3.25 | 0.27 | 0.03 | 1.15 | 0.00 | | | 2.50 | 38.1 | 4.01 | 247 | 121 |
| Egypt 1a | Turquoise (n = 9) | 17.4 | 0.70 | 3.73 | 67.1 | 0.13 | 0.96 | 0.53 | 3.72 | 0.31 | 1.19 | 1.39 | 2.04 | 0.10 | 0.27 | 3.12 | 44.1 | 7.42 | 298 | 117 |
| Egypt 1a | Purple (n = 7) | 18.0 | 0.73 | 3.79 | 67.9 | 0.15 | 0.94 | 0.60 | 4.02 | 0.31 | 1.60 | 1.51 | 0.05 | 0.00 | 0.04 | 4.23 | 41.8 | 12.8 | 340 | 127 |
| Egyptian | Green (n = 3) | 12.8 | 0.82 | 3.62 | 65.3 | 0.14 | 0.60 | 0.95 | 5.70 | 0.27 | 0.08 | 3.79 | 4.49 | 0.31 | 0.97 | 4.70 | 60.6 | 9.42 | 282 | 111 |
| Levantine | Aqua (n = 2) | 14.0 | 0.53 | 3.47 | 72.5 | 0.10 | 0.78 | 0.68 | 7.04 | 0.12 | 0.03 | 0.60 | 0.01 | 0.00 | 0.00 | 2.13 | 21.8 | 1.82 | 352 | 57.5 |
| Levantine | Green (n = 3) | 12.8 | 0.59 | 3.40 | 67.0 | 0.10 | 0.71 | 0.77 | 7.59 | 0.12 | 0.05 | 2.83 | 3.15 | 0.14 | 0.42 | 4.39 | 16.1 | 7.45 | 403 | 54.8 |

Major elements are reported as oxides (wt%) and trace elements as elements (ppm). Full analytical results are reported in S1 Table.

glass pieces (NS-W03, MR-W15, MR-W16) are similar to Egypt 1a in their sand-related elements including REE, but show differences in elements associated with the flux. For example, sodium contents are much lower (12.8% Na$_2$O compared to 17.7% Na$_2$O), while calcium contents are considerably higher (5.7% CaO compared to 3.8% CaO) compared to the Egypt 1a reference group [36]. Therefore, these samples are tentatively identified as Egyptian (Table 1), but cannot be assigned to the Egypt 1a group.

Approximately half of the tesserae ($n = 139$) and five window glass pieces that have low heavy mineral contents (TiO$_2$/Al$_2$O$_3$ < 0.05, 30–70 ppm Zr; Fig 1A and 1B) correspond to Levantine reference groups, including some Jalame, Levantine I from Apollonia, and possibly Bet Eliʿezer Levantine II type glasses [1]. Judging from their soda to silica and lime to alumina ratios (Fig 2A), the base glass of the majority of Levantine tesserae from al-Minya is consistent with either Apollonia Levantine I glass, which was discontinued in the late 7th or early 8th century CE [6, 10, 43], or glass from 4th-century Jalame [48]. Evaluating the lime, manganese,

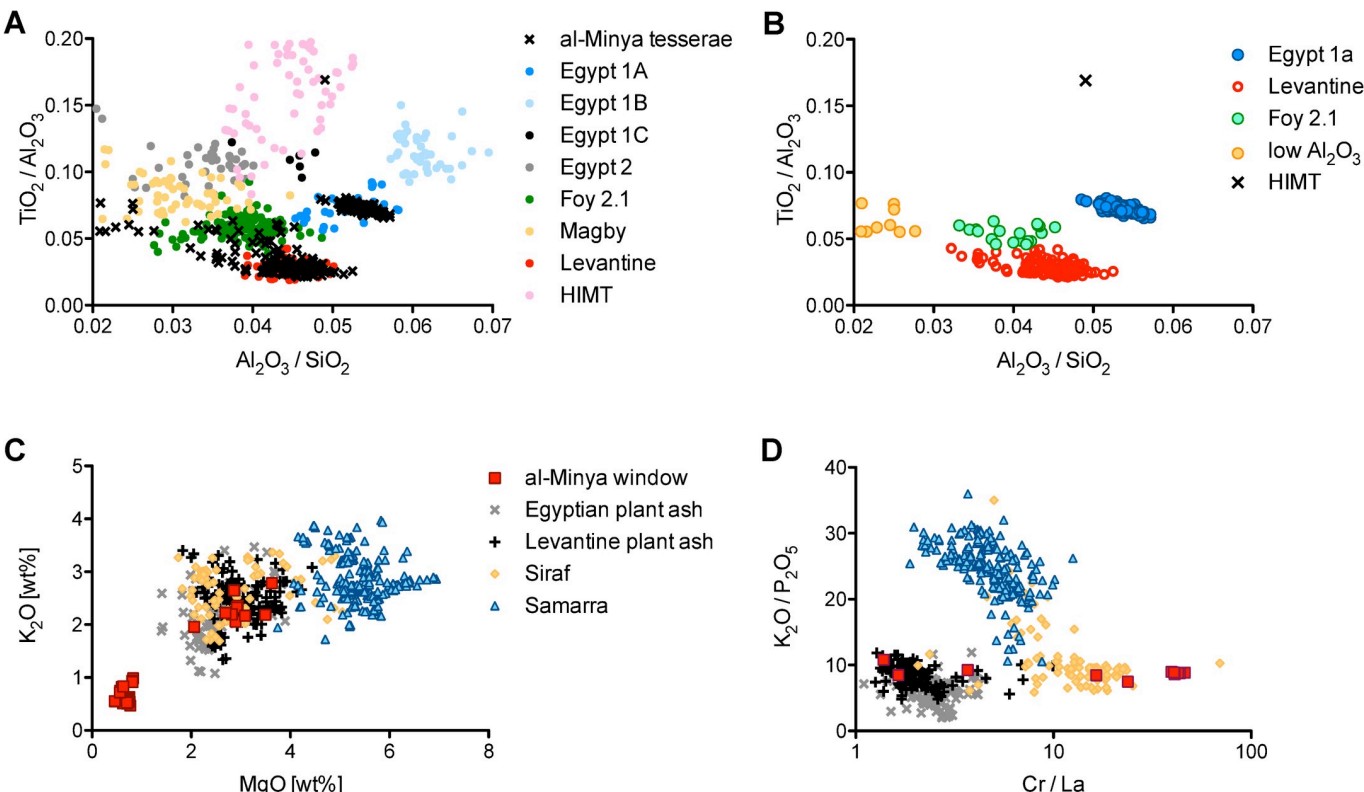

**Fig 1. Base glass characteristics of the architectural glasses from Khirbat al-Minya.** (A) $TiO_2/Al_2O_3$ and $Al_2O_3/SiO_2$ ratios of the tesserae in comparison with selected primary glass reference groups; (B) separating the tesserae according to their affiliations with primary production groups as reflected in their $TiO_2/Al_2O_3$ and $Al_2O_3/SiO_2$ ratios; (C) $K_2O$ and $MgO$ concentrations of the window glasses compared to glass reference groups distinguishing natron from plant ash glasses; (D) $K_2O/P_2O_5$ and $Cr/La$ ratios differentiate between plant ash glasses from different geographical regions. *Data sources*: [1, 10, 36–47].

potash and phosphorus concentrations, different base glasses and different degrees of recycling are apparent (Fig 2B and 2C). One set of samples has lime contents of approximately 10%, the other has on average 8% CaO. A significant number of tesserae belonging to the low calcium

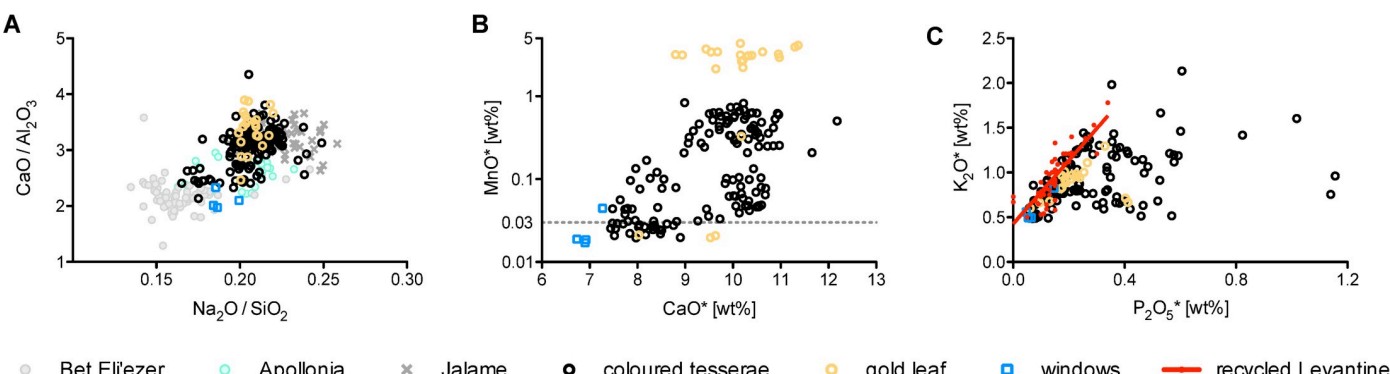

**Fig 2. Base glass characteristics of the Levantine type mosaic tesserae and window glass fragments from Khirbat al-Minya.** (A) $CaO/Al_2O_3$ versus $Na_2O/SiO_2$ ratios compared to published data from Apollonia, Jalame and Bet Eli'ezer; (B) $MnO$ and $CaO$ contents for the tesserae distinguish different base glasses and different degrees of recycling. The line at 0.03% $MnO$ indicates the proposed threshold below which manganese is considered a natural impurity of the silica source; (C) potash and phosphorus contents are positively correlated in the bulk of the samples with $P_2O_5$ levels of up to about 0.25%, indicative of re-melting and/or recycling. Higher $P_2O_5$ contents are probably due to the use of calcium phosphate as an opacifier. *Data sources*: [1, 6, 10, 51]. Asterisks indicate reduced and normalized compositions.

group has less than 0.03% manganese that is considered the threshold of natural impurities of MnO in the silica source [49]. This subset comprises the samples that resemble 8th-century Bet Eliʿezer Levantine II type glasses [1]. The remaining Levantine samples have elevated manganese contents (MnO > 0.03%), suggesting the incorporation of manganese-bearing cullet at some point during the life-cycle of these glasses or the reuse of older Roman and/or Jalame types, because Levantine I glass from Apollonia has typically low manganese contents [10, 48]. Evidence that the bulk of the Levantine glass tesserae have undergone prolonged or repeated heat treatment as part of recycling practices can be gleaned also from positively correlated potassium and phosphorus concentrations in a large fraction of the tesserae (Fig 2C) [50, 51]. For comparison, we included some data of glass artefacts from Jerash that have been demonstrated to be predominantly made of recycled Levantine glass [51]. The samples lying on or close to the regression line of the recycled Levantine vessel glass may be attributed to fuel ash contamination during recycling and/or secondary working of the vitreous material. Tesserae with even higher phosphorus contents (approximately $P_2O_5 > 0.25$) may have been opacified using calcium phosphate. This opacifying technology makes it difficult to attribute the mosaic tesserae from Khirbat al-Minya to the primary production groups of Apollonia, Jalame or Bet Eliʿezer. It is certain, however, that some older Levantine glass is present in this group of tesserae whether through recycling (re-melting), or the reuse of old tesserae.

Remnants of older glass types can also be found in the remaining glass tesserae ($n = 30$). Most ($n = 21$) resemble Foy 2.1, characterised by higher sodium, magnesium and heavy minerals such as titanium and zirconium than the Levantine group (Table 1; Fig 1A and 1B), which suggests an Egyptian origin [39, 41, 46, 49, 50]. However, it should be noted that it is difficult to establish a firm threshold between this group and the Levantine tesserae, as the distinction is obscured due to the use of colourants and opacifiers. Nine samples have very low $Al_2O_3$ and elevated MgO levels. They bear some similarities with the so-called Magby (Magnesium-rich Byzantine) glass made with a plant ash component, although the absolute MgO contents of the low-alumina al-Minya tesserae are at the lower end (1–1.5% MgO) of the range found in the literature (1–2.5% MgO) [43–46]. Also identified in the assemblage is one HIMT (high iron, manganese and titanium) tessera, a type dated to the 4th-5th century [2, 40, 41, 52–54]. All three minor groups (Foy 2.1, HIMT, low $Al_2O_3$) are thought to have been made in Egypt [40, 41, 53].

Amongst the window glass pieces are several plant ash glasses, which contain MgO and $K_2O$ contents in excess of 1.5%, generally accepted as the upper limit for natron glass production [55]. The absolute MgO concentrations of the plant ash window glass are similar to those published for plant ash glass of Levantine or Egyptian origin (~2–4% MgO; Fig 1C). However, eight of these samples, all amber/olive in colour, have elevated lithium contents (12–21 ppm) and exceptionally high chromium levels that suggest a Mesopotamian provenance (130–340 ppm, Fig 1D) [36–38, 47, 56, 57]. This is broadly similar to a recently published glass assemblage from Sīraf, Iran, which shared flux characteristics with plant ash glass from the eastern Mediterranean but whose sand-related trace elements had Mesopotamian features [47].

## 3.2 Colouring and opacifying the tesserae (secondary production)

The tesserae have been grouped broadly by colour, and include aqua/white, black, blue, gold leaf, green, olive, purple, turquoise, red and yellow. Some colours (green, olive, turquoise and yellow) are found in both the Egypt 1a and Levantine groups, while other colours are differentiated by their base glass types (Table 1; Fig 3). All of the purple tesserae are made with Egypt 1a glass, while the aqua/white, black, blue, gold leaf and red samples are all of the Levantine

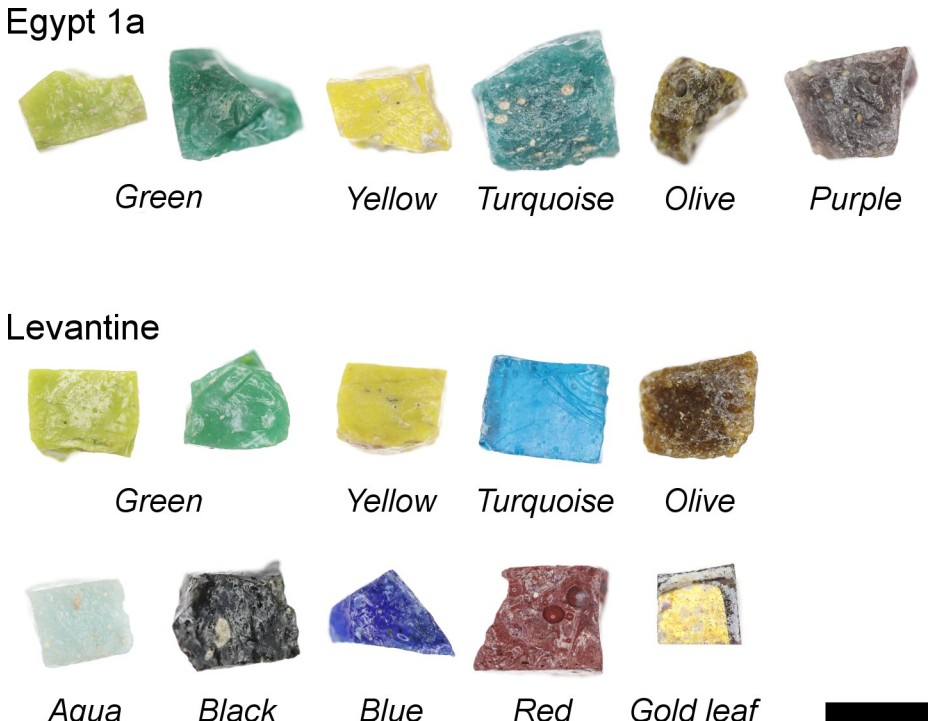

**Fig 3. Selection of glass tesserae found at Khirbat al-Minya.** The majority of the tesserae were made with Levantine or Egypt 1a primary glass. Some colours were exclusive to one type of primary glass: all the purple tesserae were made with Egypt 1a glass, while the aqua, black, blue, red and gold leaf tesserae were made with Levantine glass. Green, yellow, turquoise and olive tesserae were identified in both primary glass groups. Scale bar is 1 cm.

type. Some visual distinctions are noted macroscopically in the colours found in both base glass types (Fig 3). The Egypt 1a turquoise are opacified and greenish in hue, while the Levantine turquoise tesserae are transparent with a brilliant blue hue. The Egypt 1a green tesserae tend to be darker and more emerald in colour than the Levantine samples, although the difference is less distinctive.

Several raw materials were used to obtain different colours and opacities, in particular lead stannate, copper, calcium phosphate probably in the form of bone ash (hydroxyapatite) [14, 58], and manganese. Lead-tin oxide was added in substantial amounts to the green and yellow tesserae within both the Egypt 1a and Levantine groups. The yellow tesserae generally contain higher concentrations of lead and tin (13.4–27.5% PbO, 1.5–4.0% $SnO_2$) than the green samples (most contain 6.0–16.6% PbO and 0.6–2.0% $SnO_2$, S1 Table). Examination of the cross-section in the SEM shows the incomplete mixing of a natron glass with the lead-tin raw material (Fig 4A). Point analyses of clearly distinguishable lead-tin crystals measured a ratio of about 2.5:1 Pb:Sn, approximating the ratio of lead stannate ($Pb_2SnO_4$), although the bulk ratio is generally higher (about 10:1 Pb:Sn). The high Pb:Sn ratio favours the formation of yellow crystals of lead-tin oxide over white crystals of tin oxide, a colouring technique that is rooted in the late antique glassmaking tradition and closely linked to the early Islamic technology of white and yellow opaque glazes [59–61]. Much smaller amounts of lead-tin were added to the aqua/white, black, red and turquoise tesserae (up to about 4.5% PbO). The lead contents of the Egypt 1a tesserae show a fairly consistent ratio between PbO and Bi contents (Fig 5A). As bismuth can be used as a tracer of ore sources and metallurgical process for lead [62], this suggests a common raw material was used for the lead-rich green and yellow tesserae made with

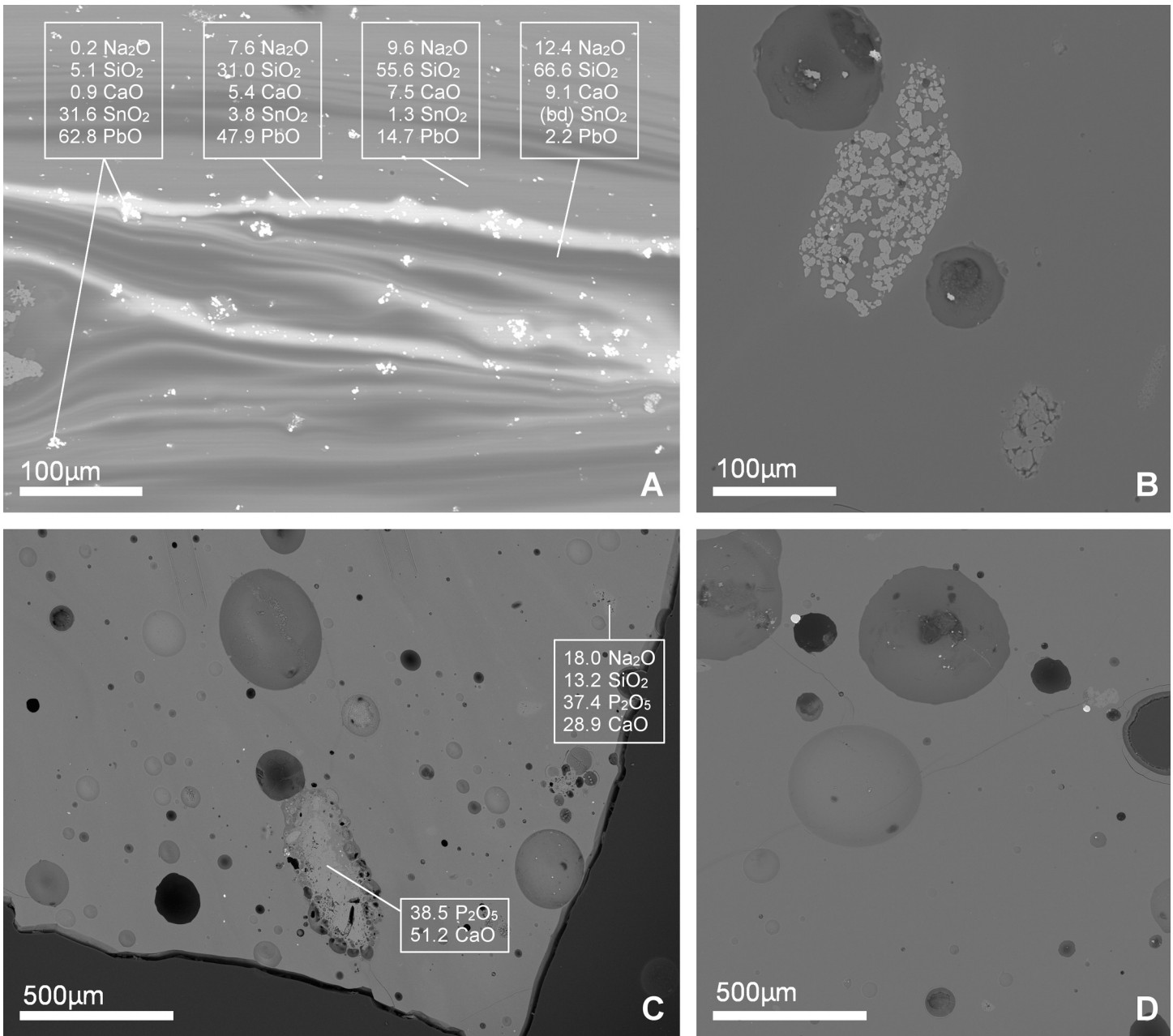

**Fig 4. Backscattered electron images of selected samples in cross-section.** (A) Yellow Levantine sample NS-T12 shows characteristic incomplete mixing of base glass and colouring agents with phases and particles of different compositions, given as means of three point analyses for selected oxides. Phases analysed are bright inclusions, the bright lead-rich phase, a mid-lead phase, and a dark lead-poor phase; (B) manganese oxide inclusions identified in purple sample NS-T40 of an Egypt 1a base glass; (C) turquoise Egypt 1a tessera NS-T30 exhibits a large inclusion of calcium phosphate, with a reaction zone and bubbles surrounding it. Also visible are small crystals of opacifier with sodium from the surrounding glass substituting for calcium in a cation exchange [58]. Tables report point analyses of each inclusion; (D) copper sulphide prills and some calcium phosphate in olive Egypt 1a tesserae (NS-T36).

Egyptian glass. Different lead-tin raw material(s), with variable ratios of PbO and Bi, were used in the Levantine tesserae, partially due to recycling and reuse of older Levantine types.

Copper was used to colour the lead-rich green as well as the turquoise (Egypt 1a and Levantine I) and the red (Levantine I) tesserae. Different redox conditions of the furnace as well as the composition of the glass affect the colours generated by copper. The cupric ion forms blue

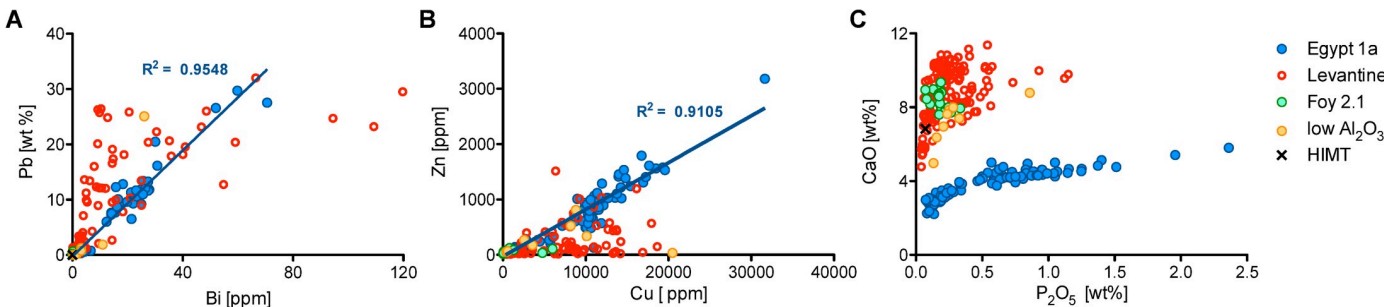

**Fig 5. Characteristics of the colouring and opacifying agents used for the tesserae from Khirbat al-Minya.** (A) Egypt 1a tesserae exhibit positive correlations between PbO and Bi that suggest a common lead raw material; (B) CuO and ZnO are positively correlated in the Egypt 1a glasses, while the Levantine glasses have more variable contents; (C) positive correlations of CaO and $P_2O_5$ in the Egypt 1a tesserae with a ratio of approximately 4:3 CaO:$P_2O_5$ is consistent with the use of hydroxyapatite $Ca_5(PO_4)_3(OH)$ as opacifying agent. A few Levantine samples have also been opacified by calcium phosphate.

in silicate glasses and green in the presence of lead oxide, while red is formed by the precipitation of dendritic cuprite particles and/or nanoparticles of metallic copper, achieved under strongly reducing conditions and possibly with subsequent heat treatment [63–66]. The red tesserae also contain elevated concentrations of iron (2–6% $Fe_2O_3$, compared to about 0.4% $Fe_2O_3$ in the other Levantine I tesserae), and elevated tin contents (up to about 0.45% $SnO_2$), which would have supported the reduction of the cupric ion and precipitation of metallic copper. The difference in hue between the Levantine I and Egypt 1a turquoise tesserae (blue versus blue-green) may be attributed in part to the higher heavy mineral content of the Egyptian glass. The higher concentrations of iron, typical of Egyptian sands, shifts the colour from blue to blue-green [67]. The copper used to colour the Levantine tesserae has variable concentrations of zinc and tin associated with it (perhaps due in part to recycling evident in the Levantine group), while the copper used to colour the Egypt 1a tesserae has a remarkably consistent ratio between the three elements, normalising to approximately 86.6% Cu, 6.6% Zn and 6.8% Sn, suggesting the use of recycled gunmetal to colour the Egypt 1a tesserae (Fig 5B; [68, 69]).

The use of calcium phosphate, probably in the form of bone or bone ash i.e. hydroxyapatite [14, 58, 70], is detected predominantly in the Egypt 1a group, where it is used to opacify the purple, turquoise, and some of the olive tesserae. These tesserae are visually opaque and the addition of calcium phosphate can be detected chemically in the bulk composition of the tesserae. CaO and $P_2O_5$ are positively correlated with a ratio of approximately 4:3 CaO:$P_2O_5$ (Fig 5C), consistent with hydroxyapatite $Ca_5(PO_4)_3(OH)$. In cross-section under the SEM, inclusions of calcium phosphate are observed, surrounded by gas bubbles and reaction zones where sodium from the surrounding glass has partially exchanged with calcium from the calcium phosphate (Fig 4C; compare to Figs in [58, 71]).

Two colour groups contain high manganese, the gold leaf tesserae (Levantine and Foy 2.1) and purple tesserae (Egypt 1a). The purple tesserae are coloured with added manganese in the range of 0.8–1.7% MnO. Mn:Fe ratios for the purple tesserae are approximately 1:1, which is surprising in light of recent experiments that suggested the most important factor for the successful generation of purple was that manganese concentrations must exceed iron contents [72]. Moreover, the manganese concentrations of most of the gold leaf tesserae exceed those of the purple glass. The Levantine gold leaf tesserae have a median concentration of about 3% MnO, while the gold leaf tesserae with a Foy 2.1 composition range from 1.3–3.8% MnO (S1 Table). The colour of the glass base of the gold leaf tesserae ranges from yellow/green to deep brown/purple, without any apparent correlation to the manganese contents. The production of colour in glass by manganese appears to be complex and is perhaps more dependent on the

redox conditions of the furnace and the oxidation states of both manganese and iron, as has been the longstanding view [63, 73, 74]. The characteristics of the gold leaf tesserae (both the base glass compositions and the manganese contents) are broadly consistent with gold leaf tesserae from other 8[th]-century contexts such as the Umayyad Mosque of Damascus and the palace at Qusayr Amra, where 'new' and re-used tesserae were identified based on their gold leaf compositions [75]. Inclusions of manganese oxide (identified using SEM-EDS) were observed in cross-section of many of the purple tesserae (Fig 4B), something not commonly reported but which has been observed previously in tesserae from the church of Petra in Jordan (where activity is dated from the mid-5[th] to early 8[th] century) [76].

The black tesserae of both the Levantine and Foy 2.1 base glasses are coloured with an iron-rich material, possibly waste products from smithing, resulting in high iron contents (2.5–9.7% $Fe_2O_3$) that are sufficient to render the glass opaque in reflected light due to the intensity of the colour and the thickness of the tesserae (Fig 3, S1 Table). In contrast, the four low alumina black tesserae were coloured with high manganese instead, and are purplish in hue when examined in transmitted light. The blue tesserae (Levantine) are coloured with cobalt and are a darker, deeper blue than the copper-coloured turquoise. Elevated concentrations of nickel, iron and lead are associated with increasing cobalt contents [77]. The cobalt to nickel ratios range typically from 2.9–8.3, with the exception of one tessera that has a significantly higher $CoO/NiO$ ratio of 17.9. The lower end of this range is consistent with the use of a cobalt colourant that emerged in the 4[th] century CE. Higher $CoO/NiO$ ratios may be linked to recycled or re-used Roman material [78].

The colour of the aqua/white tesserae (Levantine) appears to be produced variously through the addition of small amounts of lead-tin oxide, copper and/or bone ash, and/or through the manipulation of the iron content derived from the silica source with a reducing atmosphere in the furnace. The olive tesserae (both Levantine and Egypt 1a) comprise various shades of brownish, greenish or yellowish glass, and are either opaque or transparent. Three of the Egypt 1a olive tesserae have bone ash and a small amount of copper (0.03–0.04% $CuO$), resulting in a greener hue. Although the bulk copper content for these tesserae remains very low, prills of copper sulphide are observed in the SEM (Fig 4D). The colour of the tesserae may have been manipulated through furnace conditions altering the oxidation state of the iron incidental in the glass from the silica raw material.

## 3.3 Colouring the window glass (secondary production)

The vast majority of the window glasses (S1 Fig) are of the Egypt 1a type (n = 21) of different colours, while a special kind of amber/olive coloured glass (n = 8) has a plant ash composition of possibly Mesopotamian provenance. One purple and one aqua coloured plant ash glass cannot be unambiguously attributed at this point. Five samples consistent with a Levantine base glass are green (n = 3) and aqua (n = 2) coloured.

The turquoise window glass pieces (Egypt 1a) are coloured with copper associated with zinc and tin, and as with the Egypt 1a copper-coloured tesserae, the proportions between the three elements are relatively constant; in the window glasses, however, these elements normalise to 87.1% Cu, 9.0% Zn and 3.9% Sn, hence different to the proportions found in Egypt 1a tesserae. Copper was also used to colour the green glasses (Levantine: 3.1–3.2% $CuO$, and Egyptian: 4.1–4.7% $CuO$). The Egyptian green glasses comprise the three samples with sand-related elements similar to Egypt 1a compositions, but with dissimilar soda and lime contents. The copper used in these glasses is not consistent with the Cu-Zn-Sn ratios that are found in all of the other copper-coloured Egypt 1a glasses reported in this paper, neither the tesserae nor the window glass.

The amber/olive group (Mesopotamian) is characterised by elevated concentrations of iron (4.2–5.9% $Fe_2O_3$), copper (1.6–2.2% CuO) and Cr (about 300 ppm). The colour is probably owed to the presence of $Fe^{3+}$—$S^{2-}$ chromophore formed under reducing conditions [79–81]. The purple glasses (Egypt 1a and Levantine plant ash) are coloured with manganese (1.2–1.7% MnO). One of the colourless/aqua glasses (a Levantine plant ash glass) has manganese added to decolourise it (0.8% MnO). The rest have not been decolourised and have a natural tint.

## 4 Discussion

The identification of a substantial amount of Egypt 1a among the vitreous assemblage is particularly relevant for dating the palatial complex. Previously, the precise dating of the construction of Khirbat al-Minya was dependent on a building inscription that names al-Walīd as the patron, however due to the omission of a patronymic or other defining information, any distinction between al-Walīd I (705–715 CE) and al-Walīd II (743–744) could not be made [28]. The presence of significant proportions of Egypt 1a glass, the production of which has been dated to the first quarter of the 8th century [36], and the absence of other early Islamic glass types that can be attributed to after 725 CE (for example Egypt 1b, Egypt 1c, and Egypt 2), support an earlier date for the building under al-Walīd I (705–715 CE). While this conclusion is predicated on the assumption that the glass was produced and acquired contemporaneously to construction rather than stored for some decades, if the construction had taken place during al-Walīd II's reign (743–744 CE), we would expect to find some later material in the assemblage. This, for example, is the case at Khirbat al-Mafjar (dated to 736–746 AD), where the assemblage similarly comprises both Egyptian and Levantine glass but also includes the later types Egypt 1b, Egypt 1c and Egypt 2 (among the vessels) in addition to Egypt 1a, Levantine I (Apollonia) and Levantine II (Bet Eliʿezer) [13]. The attribution of the site to al-Walīd I (705–715 CE) makes Khirbat al-Minya one of the earliest known Umayyad residence buildings to date [28].

The reign of the caliph al-Walid I, considered an early zenith of the Umayyad dynasty, is characterised by numerous building works, especially the construction of numerous monumental mosques [11, 82]. Records from this period document the practice of employing craftsmen and importing materials from all over the empire. Correspondence between the Governor of Egypt and the Prefect of the district of Aphrodito (called the Aphrodito papyri) refers to the provision of workmen and materials being sent from Egypt to the Levant for the construction of the al-Aqsa Mosque in Jerusalem (706–715 CE) and the Great Mosque at Damascus (705–715 CE) [83]. According to the historian al-Balādhūrī (868 CE), al-Walīd sent from Syria and Egypt mosaic materials and craftsmen described as "Greek" and "Coptic" to the Arabian Peninsula, for the reconstruction and renovation of the mosque of the prophet at Medina (707–709 CE); later accounts also referred to mosaic cubes requested from Byzantium [83, 84]. Drawing on resources from across the empire for these monumental building campaigns would have helped to alleviate pressure on local resources at a time the Levantine glassmaking industry was undergoing a period of transition. Glassmaking activities at Apollonia waned around the end of the 7th or the beginning of the 8th century CE and the industry moved to Bet Eliʿezer, where large scale production is evidenced by multiple furnaces operating concurrently. However, there is little evidence for exportation beyond Syro-Palestine, suggesting the output of Bet Eliʿezer was mostly supplying the local market [10]. Decreasing soda contents in Levantine glass as well as an influx of imported Egyptian glass around this time have been attributed to an industry under stress [1, 10, 85], and the demands created by the intensive building works at the beginning of the 8th century likely exceeded local resources. For example, it is estimated that the Dome of the Rock alone would have required about 29

tonnes of glass for its mosaic decoration [86]. The import of Egyptian glass to Khirbat al-Minya and other sites of this period, whether in the form of raw glass that was then processed at local secondary workshops or in the form of coloured cakes or cubes, may have been necessary to meet the increase in material requirements. The reuse of older tesserae and the extensive recycling evident in the Levantine tesserae may have similarly helped extend the local supply.

Further characteristics of the Khirbat al-Minya tesserae reveal some very interesting patterns about secondary glass working practices. Notably, the differentiation of some colours by base glass type (purple made with only Egypt 1a, and aqua/white, black, blue, gold leaf and red found only in the Levantine group) and the use of different raw materials (different copper and lead sources) support the idea that the tesserae made with these two base glass types were coloured/opacified at different secondary workshops. The red, cobalt blue and gold leaf tesserae, arguably the colours that were the most expensive or most difficult to produce [86], were all made with Levantine glass and/or were older, reused tesserae. Red is widely recognised as one of the most technologically demanding colours to produce [87–90], while the production of blue and gold leaf tesserae relies upon access to a cobalt ore or the use of precious metal [75, 78, 91]. This may explain the large incidence of recycled/reused gold leaf tesserae. Elsewhere, these colours have been distinguished by their base glass composition from other colours of the same assemblage, suggesting they were sourced from a separate supplier with specialist knowledge or access to limited raw materials [61].

These findings are in contrast to the results of the tesserae from Khirbat al-Mafjar (736–746 CE), where no relationship between colour and primary glass type was discovered, and where the same colouring and opacifying technologies were used with different base glass categories [13, 14]. Despite the limited sample size of that study (16 tesserae), the authors speculated that secondary workshops may have specialised in the manufacture of tesserae of certain colours but worked with primary glass from both Egyptian and Levantine sources [14]. Indeed, both Egyptian and Levantine primary glass were found at the early Islamic workshop in Tel Aviv [92]. Although no mixing of the different base glasses was observed in Tel Aviv, such a stringent separation between Levantine and Egyptian raw glass and/or cullet is not to be expected in the production of glass tesserae. In the absence of mixed base glass compositions in a tessera assemblage, as is the case at both Khirbat al-Mafjar and Khirbat al-Minya, separate secondary workshops are a more likely scenario. What is more, the mere observation that copper and tin compounds or calcium phosphate was used independent of the base glass type [14] is not surprising and does not support a production model whereby secondary workshops produced a single colour. Instead, the use of the same colouring and opacifying raw materials for the production of chromatically different tesserae in combination with the same unadulterated base glass presents strong evidence that tesserae with different colours must have originated from the same secondary workshop. Thanks to a more rigorous approach and statistically significant number of mosaic tesserae from Khirbat al-Minya, we were able to demonstrate that multiple colours made with Egypt 1a glass probably came from a single secondary workshop (Fig 6), where the raw glass would have been coloured/opacified and formed into cakes from which the tesserae are cut. Technological commonalities within the Egypt 1a tesserae group suggest the use of the same raw materials for colouring and opacifying. The most compelling evidence for this is the correlation between copper, zinc and tin for the copper-coloured tesserae (turquoise and green) and between lead and bismuth for the lead-containing tesserae (green and yellow). These correlations suggest that a single copper alloy and lead source, were used to obtain the different copper and lead colours, thus indicating that these colours likely originate from the same secondary workshop (Fig 6). The use of bone ash almost exclusively in the Egypt 1a tesserae, specifically in the turquoise, purple and olive tesserae, may indicate a

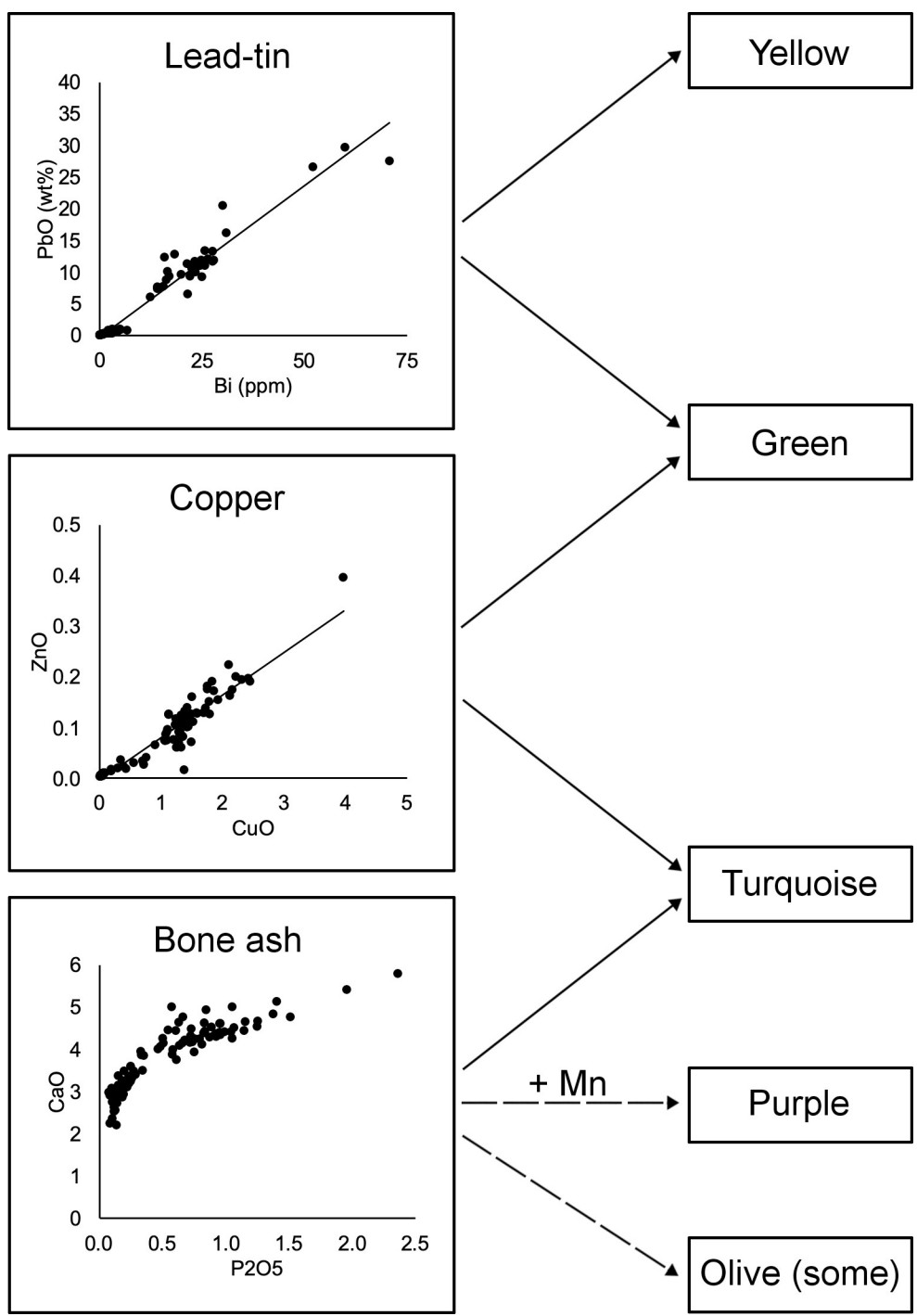

**Fig 6. Workflow diagram for the Egypt 1a tesserae found at Khirbat al-Minya.** Correlations between lead and bismuth, and zinc (and tin) and copper, suggest that the same raw materials were used to create different colours within the Egypt 1a group.

common source for these colours as well, although the use of bone ash to opacify tesserae was a widespread technology in this and preceding periods [14, 58, 71, 76].

The Egypt 1a turquoise window glass also shows a correlation between Cu, Zn and Sn, although with different ratios to the Cu-coloured tesserae. These may not be from the same

secondary workshop but are possibly from the same region with a local tradition of recycling copper alloys to colour glass. The acquisition of tesserae and window glass from separate secondary workshops may suggest specialisation between these products, although more data is required to support this hypothesis. Interestingly, there also appears to be a special source for the transparent amber/olive window glass, which are all plant ash glasses of Mesopotamian origin [93]. The amber/olive glasses are unusual, as glass with their composition (2% CuO and 5% $Fe_2O_3$) would generally be expected to be red. Red glass may be achieved through reducing the copper oxide to the colourless cuprous ion, followed by heat-treating the glass to form metallic copper nanoparticles or dendrites of cuprite [63–65, 87], although sufficiently reducing conditions may form metallic copper without heat treatment [e.g., 66]. It appears then that the amber/olive window glasses at Khirbat al-Minya were created under moderately reducing conditions that allowed for the formation of colourless $Cu^+$ but without further heat treatment that would have led to the development of dendrites or nanoparticles required to create the red colour. This is consistent with the formation of the $Fe^{3+}$—$S^{2-}$ chromophore that also requires reducing conditions within a specific range as conditions that are too reducing form the ferrous ion $Fe^{2+}$ instead of the ferric ion $Fe^{3+}$ [80, 81].

## 5 Conclusion

The chemical analyses of the window glasses and mosaic tesserae from of Khirbat al-Minya represent the largest compositional data set of Umayyad architectural glass to date, thus providing detailed information about the circulation of glass in the early 8[th] century CE. One of the key findings was the import of a large consignment of Egyptian glasses, highlighting the transitional nature of the Levantine glass industry at the time. Previously, the Levantine glass industry was largely self-sufficient, and none of the earlier Egyptian glass types has been identified in the Levant in significant quantities. The reasons for this shift are not obvious, but part of the explanation may lie in the intensity of building works during this period. The sudden and substantial increase in the demand for glass created by numerous major building campaigns is likely to have strained local supplies.

A substantial portion of the tesserae and window glass were made with Egypt 1a glass, which together with the total absence of glass types that can be dated to after 725 CE supports dating the construction of the residential building to the reign of al-Walīd I (705–715 CE) rather than al-Walīd II (743–744 CE). This would make Khirbat al-Minya one of the earliest known Umayyad residential buildings at present. Furthermore, evidence suggests that several different colours of the Egyptian tesserae were made at the same secondary workshop. In view of the unadulterated nature of the Egypt 1a base glass, secondary working of these tesserae probably took place in Egypt as well. This has not been demonstrated before, and is a significant finding for our understanding of the production model for this period, particularly in relation to strongly coloured mosaic glass, but also for the study of glass technology more generally. Finally, a special type of amber coloured window glass made with plant ash as fluxing agent appears to have been imported from Mesopotamia. The unique compositional features and distinct amber colour suggest specialised secondary production processes. To obtain the amber $Fe^{3+}$—$S^{2-}$ chromophore, a closely controlled reducing environment is required. Intriguingly, the composition of these amber glasses (2% CuO, 5% $Fe_2O_3$) is such that under strongly reducing conditions and/or with heat treatment, they may have turned out red. At this point it is not possible to decide whether the original intention was to obtain amber or whether the glasses represent a failed attempt to produce red for which the furnace conditions were not reducing enough.

## Supporting information

**S1 Fig. Selection of window glass fragments from Khirbat al-Minya sorted by colour.** Scale bar is 1cm.
(TIF)

**S1 Table. LA-ICP-MS data of the mosaic tesserae and window glass from Khirbat al-Minya.** Samples with the prefix "NS" were cross-sectioned and embedded in epoxy resin while samples with the prefix "MR" were analysed after cleaning but without further sample preparation (as described in text). Major elements are reported as oxides (wt%) and trace elements as elements (ppm). Tin and lead are reported twice, as they are major components in some glasses and traces in others. BD = below detection.
(XLSX)

**S2 Table. Average concentrations of glass standards as measured by LA-ICP-MS, compared to accepted values for Corning A, B, C, and D [32], and NIST 612 [33].** The calibration of the LA-ICP-MS data is based on the combine data of NIST610, Corning B, C and D and APL1, which are used to calculate the response coefficient factor ($K_y$) and convert signal intensities into fully quantitative data. The calculated data of Corning B, C and D can therefore still provide semi-independent measurements of precision and accuracy, especially for elements present only in low concentrations in Corning A and NIST612 such as lead, and elements that are systematically overrepresented in Corning A (e.g., calcium).
(XLSX)

## Acknowledgments

We would like to thank the Museum of Islamic Art, Berlin, and especially Jens Kröger, for allowing access to the Khirbat al-Minya assemblage.

## Author Contributions

**Conceptualization:** Laura Ware Adlington, Nadine Schibille.

**Formal analysis:** Laura Ware Adlington, Nadine Schibille.

**Funding acquisition:** Nadine Schibille.

**Investigation:** Nadine Schibille.

**Methodology:** Nadine Schibille.

**Project administration:** Nadine Schibille.

**Resources:** Markus Ritter, Nadine Schibille.

**Supervision:** Nadine Schibille.

**Validation:** Nadine Schibille.

**Visualization:** Laura Ware Adlington, Nadine Schibille.

**Writing – original draft:** Laura Ware Adlington.

**Writing – review & editing:** Laura Ware Adlington, Markus Ritter, Nadine Schibille.

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
