## [Decision Letter · Decision Letter 0]

11 Jun 2020

PONE-D-20-02692

Production and provenance of architectural glass from the Umayyad period

PLOS ONE

Dear Dr. Schibille,

Thank you for submitting your manuscript to PLOS ONE. After careful consideration, we feel that it has merit but does not fully meet PLOS ONE’s publication criteria as it currently stands. Therefore, we invite you to submit a revised version of the manuscript that addresses the points raised during the review process.

In addition to the comments from the reviewers below, we note that one of the foremost issues with this manuscript is that the authors do not do a sufficient job of demonstrating how the work expands upon current knowledge in the field. PLOS requires that studies provide new insight and that they are not merely derivative of previous work. As such, the authors should revise the Introduction to put their study in the context of previous works and to demonstrate how their study differs from and expands upon the existing body of literature. 

We look forward to receiving your revised manuscript.

Kind regards,

Natasha McDonald

Associate Editor

PLOS ONE

Journal Requirements:

2. In your manuscript, please provide additional information regarding the specimens used in your study.

Ensure that you have reported specimen numbers and complete repository information, including museum name and geographic location.

For more information on PLOS ONE's requirements for paleontology and archaeology research, see https://journals.plos.org/plosone/s/submission-guidelines#loc-paleontology-and-archaeology-research

Reviewers' comments:

Reviewer's Responses to Questions

**Comments to the Author**

1. Is the manuscript technically sound, and do the data support the conclusions?

Reviewer #1: No

Reviewer #2: Yes

Reviewer #3: Yes

2. Has the statistical analysis been performed appropriately and rigorously? 

Reviewer #1: No

Reviewer #2: Yes

Reviewer #3: Yes

3. Have the authors made all data underlying the findings in their manuscript fully available?

Reviewer #1: No

Reviewer #2: Yes

Reviewer #3: Yes

4. Is the manuscript presented in an intelligible fashion and written in standard English?

Reviewer #1: Yes

Reviewer #2: Yes

Reviewer #3: Yes

5. Review Comments to the Author

Reviewer #1: This is quite a useful compilation of data of Umayyad glass compositions from a single structure in Syro-Palestine. The interpretation based on a classification largely of natron glasses provides a provenance for the raw glass. Other recently published papers that have been cited (Fiorentino et al 2018, Fiorentino et al 2019) provide very similar results which detracts significantly from the overall novelty of the discussion and interpretations presented here. Discussion of the colorant and opacifiers use does provide some quite interesting evidence for the use of a limited source but the discussion of where that source or sources might be is very limited.

I have the following specific points that the authors need to address:

60 Excavations in the 1930s and more recent excavations by an international team have revealed evidence that there were post-Umayyad phases at Khirbat al Minya. The impression given by the authors is that the palace was a 2 phase Umayyad construction, but this is not the case. This should at the very least be discussed in detail as should its implications for the interpretation of these analyses. For example, the roof of the eastern wing was rebuilt in medieval times. Pottery assemblages date from the Umayyad, Abbassid, Crusader and Mamluk periods.

70 Why does a single slab of glass suggest that all the tesserae were produced on site? It could suggest that some were but this is an oversimplified interpretation.

113 The quality of the LAICP-MS analyses of translucent glass is not in doubt in Orleans. However, no spot size is given here and it should be. What were the levels of precision and accuracy achieved? Even if cited elsewhere good science dictates that they should be mentioned here. What about the analysis of different phases using LAICP-MS? How were inclusions located and using what imaging system? How were different crystalline inclusions analysed using LAICP-MS? What was the minimum size of crystal that could be analysed? What opaque ‘area’ was chosen as representative?

The use of an SEM is described as a means of analyzing separate phases in the glasses. How was quantification achieved? Were any standards used? What were the errors involved? Please provide these. The BSE images provided of opacifiers are useful but what about the chemical analyses carried out using the SEM? These do not appear to be used of tabulated in the paper.

126 Table not tabe.

188 It is mentioned that amber/olive ‘Mesopotamian’ plant ash glass windows were identified amongst the data. Given the good number of publications on such glasses, including some from Khirbet al-Minya, the authors should be far more specific about there the glasses are likely to have been made in ‘Mesopotamia’ and relate their findings to existing compositional evidence. A commentary on the political context of the use of such glass would also be relevant.

230 The use of a common source of colorants using Pb: Bi ratios is quite interesting, though is this a surprise? More discussion of what putative sources of colorants is needed here. Is this only related to mineralogical sources of lead or to other geological sources of colorant minerals?

249 Even if similar Pb:Bi, Zn:Sn ratios have been found in different colours in Egyptian 1a glasses why does this necessarily suggest that the same secondary workshop was involved? An alternative interpretation is that the same source provided the glass to several secondary workshops - it could have been imported from anywhere. It would be helpful to suggest some possibilities.

292 What are the Mn inclusions doing in the glass? Does this reflect a technological practice? Is it deliberate?

296 How was such a high level of Fe introduced into the glass? How does a high concentration of a transition metal produce an ‘opaque’ glass as is claimed here?

345 How can the structure be dated to the ‘production date’ of a ‘significant proportion of Egyptian 1a glass’? It is impossible to know long the period was between when the glass was fused and when it reached the site of construction. What dating technique has been used to date so precisely the manufacture of Egyptian 1a glass? This is a very shaky argument which should be toned down significantly- also please do so in the conclusion.

389 What ‘nearby suppliers’ of cobalt/gold leaf tesserae would have been used? Many publications discuss the mineralogical impurities associated with cobalt (e.g. Ni, V, Fe, Mn, As) such as those by Professor Bernard Gratuze. The authors should consider this in detail and expand and re-evaluate the discussion of the potential suppliers of e.g. cobalt glass once this is taken into account.

431 I find it difficult to believe that local supplies of glass could have been exhausted; an alternative/additional interpretation should be provided.

Reviewer #2: The subject of the article is really interesting and topical. By the way, in the last years the discussion about the primary and secondary glass production given detailed information about the acquisition and the circulation of the glass materials.

The analytical methods applied are well known, in my opinion the relevance of this paper is in the discussion and in the archaeological and historical implications.

I have one only comment:

1) Abstract. The sentence at line 29: how the authors established the oxidation state of“Fe3+—S2+ chromophore”? In the abstract, it seems a result of the paper, on the other hand at page 19 line 330 “The colour is probably owed to the presence of Fe3+—S2+ chromophore formed under reducing conditions” is a hypothesis. Please specify.

2) Fig.1C, 1D, 4 (Copper and bone ash): the axes does not have the scale: please add

Reviewer #3: File with comments has been uploaded due to the presence of figures in the text. File with comments has been uploaded due to the presence of figures in the text. File with comments has been uploaded due to the presence of figures in the text.

6. PLOS authors have the option to publish the peer review history of their article (what does this mean?). If published, this will include your full peer review and any attached files.

Reviewer #1: No

Reviewer #2: No

Reviewer #3: No

---

## [Author Response · Author response to Decision Letter 0]

24 Jul 2020

Editor Remarks to Author:

In addition to the comments from the reviewers below, we note that one of the foremost issues with this manuscript is that the authors do not do a sufficient job of demonstrating how the work expands upon current knowledge in the field. PLOS requires that studies provide new insight and that they are not merely derivative of previous work. As such, the authors should revise the Introduction to put their study in the context of previous works and to demonstrate how their study differs from and expands

upon the existing body of literature

This is the first comprehensive LA-ICP-MS study of early Islamic mosaic tesserae and window glass and the first that proposes a model of secondary production and supply, made possible due to the large number of samples. We have added some explanation in the introduction to highlight the novelty of the study and its contribution to the field and modified the conclusion to draw attention to the main historical implications. We can propose a refined dating of the palatial complex of Khirbat al-Minya, thanks to the precise dating of some of the glass finds.

Please note that we have reduced Table 1 to the major, minor and some crucial trace elements to facilitate comparison. The entire data-set is provided as supplementary material.

Reviewer #1

1. 60 Excavations in the 1930s and more recent excavations by an international team have revealed evidence that there were post-Umayyad phases at Khirbat al Minya. The impression given by the authors is that the palace was a 2 phase Umayyad construction, but this is not the case. This should at the very least be discussed in detail as should its implications for the interpretation of these analyses. For example, the roof of the eastern wing was rebuilt in medieval times. Pottery assemblages date from the Umayyad, Abbassid, Crusader and Mamluk periods.

Response: 

We have now expanded and detailed the archaeological introduction in order to include the post-Umayyad changes and use of the building, referring to various phases supported by ceramic and numismatic evidence. In addition, we have expanded the number of references to archaeological work at the site, as to make this more comprehensible (lines 56-114). 

It must be noted, however, that the medieval building of the east wing resulted not in a rebuilding of “the roof” but in a long hall over the original suite of separate rooms, as can be seen in the excavation plan of [16], is mentioned in [17] and discussed in [27]. We also wish to stress that all the tesserae analysed in this study were retrieved from Umayyad levels either set in mortar or stored in heaps. Whether stored and destined for the Umayyad decoration or collected in the building after its destruction is not entirely clear, but the material is certainly related to the Umayyad building phases in one way or other.

2. 70 Why does a single slab of glass suggest that all the tesserae were produced on site? It could suggest that some were but this is an oversimplified interpretation.

Response: 

We agree with the reviewer that this sentence was worded too strongly and we have toned it down (lines 102-106). 

113 The quality of the LAICP-MS analyses of translucent glass is not in doubt in Orleans. However, no spot size is given here and it should be. What were the levels of precision and accuracy achieved? Even if cited elsewhere good science dictates that they should be mentioned here. What about the analysis of

different phases using LAICP-MS? How were inclusions located and using what imaging system? How were different crystalline inclusions analysed using LAICP-MS? What was the minimum size of crystal that could be analysed? What opaque 'area' was chosen as representative?

The use of an SEM is described as a means of analyzing separate phases in the glasses. How was quantification achieved? Were any standards used? What were the errors involved? Please provide these. The BSE images provided of opacifiers are useful but what about the chemical analyses carried out using the SEM? These do not appear to be used of tabulated in the paper.

Response: 

The analytical protocol has now been clarified in the methods section. 

We would like to point out that LA-ICP-MS analyses were carried out on the glassy matrices and not the crystalline structures. As we specify in the methods the spot size (beam diameter) was typically set at 100 m, which had to be occasionally reduced at high Mn or Sn levels down to 50 m. Precision and accuracy are given in the supplementary material, and we have now added a few sentences to the methods section as well (lines 154-162).

Inclusions and different phases were analysed semi-quantitatively by SEM, which we have now emphasized in the text (lines 165ff). The data are only used for simple identification and for element ratios (such as between Sn and Pb or Ca and P). We have, however, added a statement about accuracy and precision of the data based on analysis of Corning A-C in response to the reviewer (lines 169-171).

126 Table not tabe.

Response: 

Done.

188 It is mentioned that amber/olive 'Mesopotamian' plant ash glass windows were identified amongst the data. Given the good number of publications on such glasses, including some from Khirbet al-Minya, the authors should be far more specific about there the glasses are likely to have been made in 'Mesopotamia' and relate their findings to existing compositional evidence. A commentary on the political context of the use of such glass would also be relevant.

Response: 

We have already drawn parallels between the al-Minya glass and published samples, in that the amber window glass at al-Minya have flux characteristics similar to Levantine/Egyptian plant ash glass, and Li and Cr contents similar to Mesopotamian glass, a combination that has a parallel in Siraf, Iran where glass with eastern Mediterranean characteristics in the flux-related elements and Mesopotamian characteristics in the sand-related elements. This is in text in lines 266-272 & Fig. 1D. However, we have included further references about Mesopotamian glass, most notably Henderson et al 2016 (the publication referred to by the reviewer, which includes some plant ash glass from al-Minya; line 256) 

230 The use of a common source of colorants using Pb: Bi ratios is quite interesting, though is this a surprise? More discussion of what putative sources of colorants is needed here. Is this only related to mineralogical sources of lead or to other geological sources of colorant minerals?

Response: 

We have now added a new Figure 5 (p. 15) that shows covariances between Pb and Bi, Cu and Zn, and Ca and P, suggesting that different colours (e.g. lead and copper coloured glass) of an Egypt 1a composition may have been produced at the same secondary workshop. While a constant Pb:Bi ratio is not surprising, it does point to the use of a common raw material for the lead additive to the glass (for the Egyptian tesserae). We have emphasized the relationship between Bi and Pb ores and metallurgical practices by adding a sentence and a reference in text (line 305ff). In the discussion, we bring this together with other results (eg the covariance of Zn, Sn and Cu) to support our argument of a common workshop. 

249 Even if similar Pb:Bi, Zn:Sn ratios have been found in different colours in Egyptian 1a glasses why does this necessarily suggest that the same secondary workshop was involved? An alternative interpretation is that the same source provided the glass to several secondary workshops - it could have been imported from anywhere. It would be helpful to suggest some possibilities.

Response: 

The argument of the same secondary workshop rests upon the interrelated covariances of different colours (Fig. 6), meaning that while the yellow and green samples share in the same lead source, the green in turn share in the same copper source as the turquoise samples, that in turn share some opacifier characteristics with the purple and olive tesserae. 

Regarding the reviewer’s alternative interpretation, the current model of production for this period as we outline in the first paragraph of the paper is that raw glass chunks were distributed to secondary workshops where the colouring and shaping took place. We have added a sentence in the discussion to re-emphasise this point for clarity (line 478ff). Under this model, it is highly unlikely that unconnected secondary workshops would receive the same raw materials (both base glass and colourants) which then ended up at the same site to be used together in wall mosaics. Along similar lines, if the secondary workshops were not in Egypt, substantial admixture of non-Egyptian glass types may be expected (see modifications in the conclusion).

292 What are the Mn inclusions doing in the glass? Does this reflect a technological practice? Is it deliberate?

Response: 

As stated in the text, we have only been able to find one other mention of Mn inclusions in glass. This is not enough evidence to speculate about technological practices, and our study is not focused on this one possibly uncommon phenomenon that would probably require experimental melts in order to understand it fully. We do not expect this to be deliberate but rather the result of the colouring process. However, we wish to report it in case there are other, unreported instances of this and in case it is addressed in future work.

296 How was such a high level of Fe introduced into the glass? How does a high concentration of a transition metal produce an 'opaque' glass as is claimed here?

Response:

We agree that our language here was not specific enough and we have adjusted it accordingly (lines 375ff). 

345 How can the structure be dated to the 'production date' of a 'significant proportion of Egyptian 1a glass'? It is impossible to know long the period was between when the glass was fused and when it reached the site of construction. What dating technique has been used to date so precisely the manufacture of Egyptian 1a glass? This is a very shaky argument which should be toned down significantly- also please do so in the conclusion.

Response: 

The dating of the compositional group Egypt 1A is based on the analysis of a substantial number of Egyptian glass weights some of which can be precisely dated due to the names of the officials inscribed on the objects. No Egypt 1A post-dates the year 725 CE, by that time this compositional group had been replaced by a new (albeit related) Egypt 1B composition. Hence, no material that can unambiguously attributed to a date later than 725 CE was identified among the finds from al-Minya, with the possible exception of some plant ash glass that have been exclusively identified among the window glass fragments and some vessel glass (e.g. Henderson et al., 2016).

The reviewer is correct that we do not know when exactly the glass was fused, but we can say with some certainty that Egypt 1A was not produced after the first quarter of the 8th century. Concomitant with the lack of later material, we conclude that an earlier date for the construction of al-Minya is more likely. If the palace were constructed during the reign of al-Walid II (743-744), at least some later material would be expected, similar to the assemblage from Khirbat al-Mafjar. The large number of samples analysed in our study implies that our data set can be considered representative of the entire assemblage from al-Minya.

We clarified the rational in the text (lines 422ff) and the conclusion (lines 533ff).

389 What 'nearby suppliers' of cobalt/gold leaf tesserae would have been used? Many publications discuss the mineralogical impurities associated with cobalt (e.g. Ni, V, Fe, Mn, As) such as those by Professor Bernard Gratuze. The authors should consider this in detail and expand and re-evaluate the discussion of the potential suppliers of e.g. cobalt glass once this is taken into account.

Response: 

The mineralogical impurities associated with cobalt, such as Ni, As, Fe, Pb, give evidence about the cobalt ore, not about the location of secondary production. The data merely suggest that the cobalt-coloured samples all correspond to Levantine base glasses and were thus from the region. We have extended our reporting of the cobalt characteristics (lines 381ff) and it has proved useful for our added material on recycling instigated by Reviewer 3.

431 I find it difficult to believe that local supplies of glass could have been exhausted; an alternative/additional interpretation should be provided.

Response: 

There can be no doubt that Levantine primary natron-type glassmaking was in decline in the eighth century, reflected in decreasing soda levels as well as in sizeable imports of Egyptian glass for domestic use and monumental architectural campaigns alike (see e.g. Phelps et al., 2016). This is significant, because up to this point, the Levantine glass industry appears to have been virtually self-sufficient. Prior to the 8th century, the late antique Egyptian glass groups that were widespread throughout the Mediterranean are only rarely found among Levantine assemblages. Furthermore, the amount of glass required for some of the monumental architectural campaigns (e.g. Great Mosque of Damascus) was enormous that most certainly put a strain on the Levantine glass industry.

We have added a disclaimer with references to strengthen our argument in the discussion (lines 450ff) and rephrased our statement in the conclusion (line 525). 

Reviewer #2

1) Abstract. The sentence at line 29: how the authors established the oxidation state of"Fe3+--S2+ chromophore"? In the abstract, it seems a result of the paper, on the other hand at page 19 line 330 "The colour is probably owed to the presence of Fe3+-- S2+ chromophore formed under reducing conditions" is a hypothesis. Please specify.

Response: 

The reviewer is correct, the statement in the abstract is somewhat misleading and was thus removed especially because this is only a relatively minor detail of the overall findings of the paper.

2) Fig.1C, 1D, 4 (Copper and bone ash): the axes does not have the scale: please add

Response: 

New figures have been prepared.

Reviewer #3

1. The sample classification needs to be more detailed and include discussion of the possible presence of Egypt 1c.

2. There has to be a detailed section that look at recycling. There is no such discussion. Instead, it is ruled out based on one tesserae block found in the complex. It has to be dealt with – in particular the Levantine samples show clear indications of recycling and presence of pre-Apollonia glass such at that produced at Jalame.

3. Small, but important. There are not enough references to figures.

See individual comments below.

Author list:

What were author contributions? Since last author is corresponding author, is the order alphabetical?

Response: 

We now included author contributions. 

Introduction:

Tesserae are found 1. as loose and embedded in mosaics with mortar bedding and 2. Without mortar in a large deposit. ‘Additionally, a fragment of a glass slab in the thickness of the tesserae was found with marks from impact …….., suggesting this was the material from which the tesserae were cut on site. This indicate that some, if not all, of the glass are contemporary to the site.’ This is a critical statement to the paper since they use it to conclude that all Egypt 1 and Levantine glass is primary and contemporaneous with the complex. Based on this, they conclude that the complex was build in AD 705-715 rather than 743-744. – Yet there is no description of the glass slab: What was the glass type? What was the colour?

Why are they convinced that the glass is contemporary to the site?

Why could it not be recycled translucent glass that was melted, coloured and shaped into the slab for tesserae production? If so, it could be older glass. There needs to be a discussion on recycling rather than just dismissing it on the basis of one glass slab.

Response: 

We agree with the reviewer that this sentence was worded too strongly and we have toned it down (lines 102-106). Especially, we have removed the interpretation that the material is contemporary to the construction of the palace. However, it was only ever our intention to report an interesting and relevant find, and the arguments we put forward in this paper are not founded upon this one find. 

We can claim with confidence that Egypt 1a was not produced after the first quarter of the 8th century (subject of a previous paper by Schibille et al 2019). Concomitant with the lack of later material (no Egypt 1B and no Egypt 1C), we conclude that an earlier date for the construction of al-Minya is more likely. If the palace was constructed during the reign of al-Walid II (743-744), we would expect to find at least some later material, similar to Khirbat al-Mafjar. The large number of samples analysed in our study implies that our data set can be considered representative of the entire assemblage from al-Minya.

The last paragraph of the introduction states the purposes or rather the main observations (?) of the study, but is very vague. If the authors want to summarize their observations in the introduction, they should be specific.

Purpose or observation:

1. ‘Using chemical compositions of glass from Khirbat al-Minya for an in-depth examination of the acquisition and circulation of glass during early 8th century’. Bold statement when the paper deals with one site only. Be specific – what was learned about the glass acquisition and circulation?

2. “Multiple mosaic colours appear to have been made at the same secondary workshop, yielding insights into the production model for secondary glass working’. Again, grand statement – but be specific. What insights did it yield?

Response: 

We have now rephrased the last paragraph of the introduction to highlight the novelty and main contribution of our study. 

Methods:

A total of 270 glass tesserae and 37 window glasses were examined. Of there, 210 tess and 23 window glasses were analyzed by LA-ICPMS without sample prep (NS), while 60 tess and 14 window glasses (MR) were mounted in epoxy, polished and studied by LA-ICPMS, scanning electron microscope SEM-EDS. 

Why not do some repeats of the same sample by both methods for comparison to test the differences?

Response:

Since LA-ICP-MS involves a pre-ablation of surface alterations, there is no difference between prepared and unprepared samples. 

The authors use the Corning B, C and D standards for calibration, but then also for reproducibility / accuracy determinations. That does not make sense. The point of running external standards as unknowns is to choose some that would fall elsewhere on the calibrations curves – the same as unknown samples. 

I would strongly encourage that they only report the standards that were not used for calibration (Corning A and NIST SRM 612).

Line 109. Give some idea of the obtained reproducibilities and accuracies are. Just a couple of

lines saying that the reproducibility was generally x % for elements above x ppm and x % for

elements below x %. And do the same for accuracy

Response: 

We agree that including standard material used for calibration is problematic because these standards cannot be considered fully independent. However, in our case the calibration is done on the basis of the combined data of NIST610, Corning B, C and D and APL1 that are used to calculate the response coefficient factor (Ky) and convert signal intensities into fully quantitative data. Hence, the calculated data of Corning B, C and D can still provide some semi-independent measurements of precision and accuracy, especially for those elements that are only present at low concentrations in Corning A and NIST612 such as lead, and elements that are systematically overrepresented in Corning A (e.g. calcium). Furthermore, these measurements allow us to control the stability of the measurements throughout the analytical run. We would therefore like to leave these data in table S2, but if the reviewer and/or editor insist, we are happy to take them out. For transparency, we have edited the caption text for Table S2 to include parts of the above (lines 816ff).

We have also included a statement regarding precision and accuracy in the methods section as advised (lines 153ff).

Results:

Egypt1a - line 146-151 only 5 lines

The authors state that ‘one third of tesserae and half of the window glass samples classify as

Egypt 1a dated to first quarter of the 8th century’. They base this on Al2O3/SiO2 > 0.05, TiO/Al2O3 = 0.07 and 70-120 ppm Zr (line 151) and refer to Fig. 1A. Yes, figure 1A implies that these are Egypt 1 type glasses, but not that they belong to subgroup Egypt 1a. 

The authors need to present arguments for why these samples are Egypt 1a (e.g. Sr/Ca, Zr/Th, La/TiO2 etc. Schibilles own work 2019). In fact, samples MR-T100, 103, 107 and 108 have Sr/Ca ratios around 0.013, which relate them to Egypt 1c and not Egypt1a according to Schibille et al., Fig. 1b. This should be addressed in the text.

Response: 

Zr/Th and La/TiO2 does not usually distinguish between Egypt 1A and 1C, while the ratio of strontium to lime is only reliable when the two elements were introduced only with the sand source and not with the colouring and/or opacifying agents. Samples MR-T100, 103, 107 and 108, contain significant amounts of manganese that appears to have augmented the strontium levels, which has been observed by Freestone in the past. Hence, the tesserae can be attributed to Egypt 1A with confidence. Furthermore, Al2O3/SiO2 and TiO2/Al2O3 can be used to distinguish Egypt 1a, 1b and 1c - we added a graph with reference material to illustrate their attribution (now Fig 1A).

Egypt1a sand source - line 151-158

Line 151. ‘3 samples, …., are similar to Egypt 1a in their sand-related elements including REE’. Please elaborate on this statement and why not show this in a figure? Is it even possible to use REE and sand-related elements or ratios to say these coloured samples have the same sandsource as Egypt 1a? Or is it more correct to say they have the same source as Egypt1?

Are REE patterns and other sand-related elements for Egypt 1a very characteristic compared to other natron groups and even other Egypt1 groups? From figure 2 in Schibille et al., 2019, it looks like Egypt 1a,b,c have very similar REE patterns. Since the authors are not using reduced compositions, it would be hard to differentiate between REE patterns for Egypt 1a,b,c since it looks like the ratios between the REE are similar.

Response: 

The reviewer is right to question REE patterns of natron-type glasses. As has been shown previously, the REE patterns are very similar across a wide range of natron type glasses and heavy elements and ratios may be more efficacious in distinguishing between primary production groups. 

We did not include a graph showing the trace elements partially because tesserae contain substantial additives that may impact the absolute concentrations of trace elements and this is why we ‘generally favour element ratios’ as we state in our paper.

Levantine - line 170-175 only 5 lines

The authors state that half of tesserae and five window glass samples classify as Levantine with TiO2/Al2O3 < 0.05. Figure 1A should be referenced here and not just in the beginning of the result section. They go on to classify the Levantine samples according to their Na2O/SiO2 ratios and CaO/Al2O3 ratios (Phelps et al., 2016) and conclude that they are Levatine 1 type glasses produced at Apollonia. I disagree that it is this simple (see examples below).

Above is a plot that shows PbO versus CaO for the yellow tesserae of Levantine type. Here, it looks like Pb-tin colourant/opacifyer was mixed with two distinct types of Lavantine base glass characterized by different CaO concentrations. If the samples are sorted according to CaO concentrations, it is clear that the samples with high CaO also have MnO concentrations above 0.02.

Thus, while group 1 with low CaO could be from Apollonia, this cannot be the case for group 2 since base glass from Apollonia does not contain MnO. It is more likely a component of older Levantine glass from Jalame in the high CaO group.

The paper doesn’t consider the possibility of recycling, but looking solely at the yellow tesserae, there is a correlation between K and P that is independent of Pb (and thus Pb-tin addition). See two plots for reported composition and reduced compositions. This indicates that these glasses are made of recycled Levantine base glass contaminated by K and P during the remelting/recycling.

Similar considerations should be done for the other coloured groups. For the individual groups, the Ca/Al2O3 ratio (Phelps et al., 2016) is higher in samples with high MnO (or MnO/SiO2) – again, a sign that there is older byzantine (Jalame) glass present in the Levantine base glass

Response: 

Thank you to the reviewer for pointing out the differences among the Levantine samples. We have added a paragraph about the differences in the base glass and different degrees of recycling, illustrating the characteristics by means of 2 new graphs (Fig. 2B, C). 

We would like to point out that this does not change our overall interpretation of the assemblage of being contemporary or pre-dating the building of the palace of al-Minya.

Colouring and opacifying

This is great, well-written section. I miss a comparison of the gold-leaf tesserae to the observations by Neri et al., 2016 (Glass and gold) – even though this paper is referenced, it is not discussed and the materials not compared. Again for the gold leaf tesserae, it looks like there is a significant recycling component (see K-P correlation below) – and there could therefore be significant pre-Apollonia material in the Levantine gold leaf tess, but Mn cannot be used to fingerprint this since it is used for colourant

in these.

Response: 

We have now added text comparing our data with those of Neri et al. (lines 366ff) however we wish to point out that the main focus and conclusions of that paper was on dating the manufacture of gold leaf tesserae based on the composition of the gold leaf, not the glass, which they generally classified as Levantine natron, and containing Mn and/or Sb.

Line 266. ‘The copper used to colour the Levantine tesserae have variable concentrations of zinc and tin’. Could this also be a reflection of the recycling?

Response:

Yes, that is possible and we have added some text to this effect (line 341ff).

Discussion

Line 382. Note that the Levantine glass used for gold leaf tesserae also could be older and reused (see above).

Response:

We have substantially modified the paragraph on Levantine glasses, including issues of recycling supported by new graphs.

Line 392. Figure 4 is a critical figure in the text and should therefore be described and introduced in details. ‘Figure 4 presents …..’ rather than referred to once and then not again in the next paragraph although this is discussing the figure.

Response:

We agree, original Figure 4 was not sufficiently introduced and discussed. We have now added an additional figure 5 to the results section showing the covariances between Pb and Bi, Cu and Zn and Ca and P, while we discuss the full implication of figure 6 in the Discussion section.

Conclusion

Line 433-434. ‘Red, cobalt blue and gold leaf are made from primary Levantine glass’. Again, the authors need to address the recycled nature of the Levantine glass.

Response:

We have amended this sentence (lines 527ff).

Minor comments:

Line 117. ‘No permits were required for the described study, which complied with all relevant

regulations’. This sentence is out of context and belongs with sample selection.

This sentenced was moved to the sample selection (now line 122).

Line 126. Typing error ‘Tabe’ = Table

Done

Line 163. ‘CaO/Al2O3 versus Na2O/SiO2 contents’ should be changed to CaO/Al2O3 versus

Na2O/SiO2 ratios.

Done

Line 171. Levantine samples with TiO2/Al2O < 0.05 – refer to figure 1A here.

Done

Line 230. Bi & Pb correlation for Egypt 1 samples. Refer to plot in figure 4.

A new Figure 5 was added and referred to throughout the relevant passages.

Line 268. Provide a reference for gunmetal composition.

References to Ponting 2008 and Dungworth 1997 were added (line 344).

Line 274. Correlation CaO and P2O5 showing bone ash, again refer to plot in Figure 4 where this is shown.

Done.

Line 279. ‘The gold leaf tesserae (Levantine)’, I think it should be The gold leaf tesserae (Levantine & Foy 2.1).

Yes, this is correct, we have changed it accordingly.

Line 322. Copper also underlies the green glasses. Confusing sentence.

Sentence was re-phrased.

Figures:

Figure 1 is very smudged so it is hard to see the details.

Response: We assume that high resolution figures will be made available to the reviewers by PloSOne.

---

## [Editor Report · Decision Letter 1]

26 Aug 2020

PONE-D-20-02692R1

Production and provenance of architectural glass from the Umayyad period

PLOS ONE

Dear Dr. Schibille,

Thank you for submitting your manuscript to PLOS ONE. After careful consideration, we feel that it has merit but does not fully meet PLOS ONE’s publication criteria as it currently stands. Therefore, we invite you to submit a revised version of the manuscript that addresses the points raised during the review process.

You and your colleagues have responded in detail to the reviewers comments on the archaeological context, different aspects of glass technology, the chemical characterisation of the glasses, their likely provenance and the relationship to models of production and distribution. You have done this to the extent that the article is now very close to being acceptable for publication. 

The only further improvement that needs to be made in order for it to be publishable is to refer to Fiorentino et al's publications on similar material in AAS 2018/ 2019 more extensively and in much more detail.

We look forward to receiving your revised manuscript.

Kind regards,

Julian Henderson, PhD

Academic Editor

PLOS ONE

---

## [Author Response · Author response to Decision Letter 1]

4 Sep 2020

… to refer to Fiorentino et al’s publications on similar material in AAS 2018/2019 more extensively and in much more detail.

We have expanded and clarified the base glass diversity (lines 436-439) and added a paragraph on the secondary production model based on the colouring and opacifying materials (lines 487-507), proposing a different scenario for the organization of secondary workshops dedicated to the manufacture of mosaic tesserae.

---

## [Editor Report · Decision Letter 2]

14 Sep 2020

Production and provenance of architectural glass from the Umayyad period

PONE-D-20-02692R2

Dear Dr. Adlington,

We’re pleased to inform you that your manuscript has been judged scientifically suitable for publication and will be formally accepted for publication once it meets all outstanding technical requirements.

Kind regards,

Julian Henderson, PhD

Guest Editor

PLOS ONE

Additional Editor Comments (optional):

Thank you for your modifications and changes to the paper. It is now publishable.
---

## [Editor Report · Acceptance letter]

18 Sep 2020

PONE-D-20-02692R2 

Production and provenance of architectural glass from the Umayyad period 

Dear Dr. Schibille:

I'm pleased to inform you that your manuscript has been deemed suitable for publication in PLOS ONE. Congratulations! Your manuscript is now with our production department. 

Kind regards, 

on behalf of

Dr. Julian Henderson 

Guest Editor

PLOS ONE